# Reporting on the Value of Artificial Intelligence in Predicting the Optimal Embryo for Transfer: A Systematic Review including Data Synthesis

**DOI:** 10.3390/biomedicines10030697

**Published:** 2022-03-17

**Authors:** Konstantinos Sfakianoudis, Evangelos Maziotis, Sokratis Grigoriadis, Agni Pantou, Georgia Kokkini, Anna Trypidi, Polina Giannelou, Athanasios Zikopoulos, Irene Angeli, Terpsithea Vaxevanoglou, Konstantinos Pantos, Mara Simopoulou

**Affiliations:** 1Centre for Human Reproduction, Genesis Athens Clinic, 14-16 Papanikoli, 15232 Athens, Greece; sfakianosc@yahoo.gr (K.S.); agni.pantos@gmail.com (A.P.); lina.giannelou@gmail.com (P.G.); renaangelis@yahoo.co.uk (I.A.); t.vaxevanoglou@hotmail.com (T.V.); info@pantos.gr (K.P.); 2Department of Physiology, Medical School, National and Kapodistrian University of Athens, 75 Mikras Asias, 11527 Athens, Greece; vagmaziotis@med.uoa.gr (E.M.); sokgrigor@med.uoa.gr (S.G.); gkokkini@med.uoa.gr (G.K.); atrypidi@med.uoa.gr (A.T.); 3Obstetrics and Gynaecology Royal Cornwall Hospital, Treliske, Truro TR1 3LQ, UK; thanzik92@gmail.com

**Keywords:** artificial intelligence, IVF, data-synthesis

## Abstract

Artificial intelligence (AI) has been gaining support in the field of in vitro fertilization (IVF). Despite the promising existing data, AI cannot yet claim gold-standard status, which serves as the rationale for this study. This systematic review and data synthesis aims to evaluate and report on the predictive capabilities of AI-based prediction models regarding IVF outcome. The study has been registered in PROSPERO (CRD42021242097). Following a systematic search of the literature in Pubmed/Medline, Embase, and Cochrane Central Library, 18 studies were identified as eligible for inclusion. Regarding live-birth, the Area Under the Curve (AUC) of the Summary Receiver Operating Characteristics (SROC) was 0.905, while the partial AUC (pAUC) was 0.755. The Observed: Expected ratio was 1.12 (95%CI: 0.26–2.37; 95%PI: 0.02–6.54). Regarding clinical pregnancy with fetal heartbeat, the AUC of the SROC was 0.722, while the pAUC was 0.774. The O:E ratio was 0.77 (95%CI: 0.54–1.05; 95%PI: 0.21–1.62). According to this data synthesis, the majority of the AI-based prediction models are successful in accurately predicting the IVF outcome regarding live birth, clinical pregnancy, clinical pregnancy with fetal heartbeat, and ploidy status. This review attempted to compare between AI and human prediction capabilities, and although studies do not allow for a meta-analysis, this systematic review indicates that the AI-based prediction models perform rather similarly to the embryologists’ evaluations. While AI models appear marginally more effective, they still have some way to go before they can claim to significantly surpass the clinical embryologists’ predictive competence.

## 1. Introduction

The reduction in fertility rates in recent years has led to increased implementation of assisted reproduction techniques (ART) [1]. Outcomes of in vitro fertilization (IVF) depend on multiple parameters and their respective intertwined associations. Undoubtedly, a major determining factor for a successful IVF outcome is embryo quality [2]. A number of national and international societies, associations, and committees on reproductive medicine have proposed standardized criteria for embryo grading [3,4]. Despite the abundance of proposed embryo grading and evaluation systems buttressed by thorough validation, a consensus has not yet been reached, and research is ongoing to indicate an accurate and universally applied method [5]. Although the majority of grading systems are detailed and employ a series of morphological parameters, the process of embryo evaluation involves the interpretation and application of these criteria by the clinical embryologist and their respective individualized evaluation [5,6].

Embryo selection mainly depends on developmental rate and morphological assessment, employing light microscopy [3]. The conventional manual assessment of embryo morphology has been observed to present a significant degree of inter- and intra-observer variability. Additionally, the cut-off values regarding the morphological findings are not clearly defined, which may increase inter-observer variation regarding embryo grading [7]. The anticipated lack of consistency in grading inevitably leads to inconsistency in decision-making. Furthermore, embryo development is a dynamic process. Hence, embryo classification may vary between different observation times. This renders an all-inclusive embryo evaluation reporting on the true identity of the embryo challenging when relying on static morphological evaluation [8].

Time-lapse technology is a noninvasive method that includes sustained monitoring of embryo development employing simple algorithms that assess the timing of developmental milestones. This enables the assessment of the morphokinetics of the embryo. In this context, automated prediction models may be crucial in order to enhance objectivity and reliability of embryo grading [9]. Despite its promising nature, there is insufficient evidence to support that the advanced method of time-lapse microscopy (TLM) provides improved results compared to conventional methods for human embryo selection [10]. The majority of machine learning algorithms developed for embryo assessment require user-defined input parameters. Hence, their predictions depend on currently existing classifications, limiting further outcome improvement [6,11].

Artificial intelligence (AI) is a broad term including specific sub-areas, such as artificial neural networks (ANNs), deep learning, and machine learning [12]. AI prediction models are defined as the development of algorithms, employing big data, with the ability to learn and display intelligent behavior. Artificial neural networks (ANNs) are the most commonly employed method of AI [13]. There are several types of ANNs, which are defined by the architecture of the layers. The common type of ANN is the convolutional neural network (CNN) employing spatial and configuration information that processes two dimensional (2D) or 3D images as input. Its architecture enables a reduction in the number of parameters and computations involved in the network [14,15]. Recurrent neural networks (RNN) are designed to define temporal or sequential information. RNNs gathering information from other data points in a sequence can reliably predict the outcome [16]. ANN models can process microscopy images or time-lapse videos as input to predict embryos’ implantation potential. The type of input is defined by different algorithms and software implementations [13].

A support vector machine (SVM) is a distance-based binary classifier. This model performs a nonlinear mapping from the input data, thus classifying and separating samples into two or more categories and transforming categorical variables into numeric form [12,17]. In IVF, SVM employs both clinical characteristics and embryo morphology to predict IVF outcome [12]. Deep learning models simulate the biological nervous system, as information is processed through interconnecting neurons presenting with numerous levels of deep layers [18]. Deep learning employs a back-propagation algorithm, allowing the AI to discover a pattern in a large dataset and to conduct alterations in its parameters. Deep learning algorithms are designed to study time-lapse video without the employment of annotated parameters and are able to represent datasets optimally [19,20]. It should be noted that different designs of AI should be available in order to provide benefits depending on each laboratory’s standard operation protocols (SOPs), equipment, and available computational power. Besides the cost that may burden IVF laboratories, training of embryologists is a necessity, and such provisions should be made for training as well as for the anticipated learning curve. It is of importance for the clinical embryologists not only to be able to interpret the results of the prediction models but equally to be able to understand whether the model may need to be retrained to adapt to a change in the SOPs. If a prediction model is trained in different laboratory culture conditions than the ones it is employed on, then it may be possible that the model underperforms. However, it should be noted that, despite these concerns, it has been reported that novel models may be able to perform similarly in different clinics that may employ slightly different SOPs [21]. Could this become a case of one size fits all? Further data will delineate this matter on whether standing prediction models can in fact be successfully employed in various clinical settings, whether a new model should emerge as capable of performing in any setting, or whether different models should be developed custom-made to perform according to specific laboratory settings, SOPs, and computational power.

In the field of IVF, the employment of AI is steadily gaining support. A number of studies investigating the effectiveness of innovative AI systems have been conducted. Several AI-based prediction models have been developed, documenting successful identification of embryos with the highest implantation potential [21,22]. Interestingly, studies, hitherto, have presented conflicting results. To elaborate on that, the predictive value of Tran’s IVY AI model has been categorized as excellent, while Geller’s AI model achieved slightly lower accuracy than conventional embryo selection [21,23]. The controversy documented, along with the variation in outcome reporting between studies, has served as the driver prompting the current data synthesis to be conducted. This contribution aims to provide a much-needed, all-inclusive evaluation aspiring for precision when reporting on the predictive strength of AI. The increased number of data and subsequent statistical power renders a meta-analytical approach highly significant [24]. This study uniquely brings to literature an analysis of the effectiveness of AI in accurately predicting the embryos with the highest implantation potential in IVF.

## 2. Materials and Methods

### 2.1. Search Strategy

A systematic search of the literature was performed in databases of Pubmed/Medline, Embase, and Cochrane Central Library on 6 April 2021. An update was performed on 9 August 2021. To cover the study questions the following keywords along with combinations of them were employed: “In vitro fertilization”, “intracytoplasmic sperm injection”, “assisted reproduction”, “artificial intelligence”, “machine learning”, “deep learning”, “support vector machine”, and “neural network”. The initial search yielded 694 studies from the three databases. Following removal of both duplicate studies (*n* = 97) and reviews, case series, case reports, commentaries, letters, and editorials, a primary study selection for detecting relevant articles was performed based on title and abstract screening, as depicted in the flowchart of Preferred Reporting Items for Systematic Reviews and Meta-analysis (PRISMA) (Figure 1). Screening and selection of the relevant literature was performed by two independent authors. Any discrepancies between the authors were resolved by an arbitration mediated by the senior authors. Identification of other eligible studies was performed by forward and backward citation mining on the selected relevant literature employing Google Scholar. The screening process resulted in a total of 18 studies that were eligible to be included in the present systematic review and data-synthesis. The study has been registered in PROSPERO (CRD42021242097).

### 2.2. Study Selection

In the present study, the authors opted to include both prospective and retrospective cohort studies that were performed only in humans and published in English. The population comprised preimplantation embryos in the laboratory selected for embryo transfer for women undergoing IVF/ICSI cycles. In order for a study to be eligible for inclusion, the embryo selection for transfer needed to have been performed with the employment of an AI-based prediction model.

### 2.3. Excluded Studies

One study was excluded; although it was probably relevant to our study’s scope, as its publication language was Polish [25]. Additionally, studies that focused on the prediction of blastocyst quality with an AI-based model but did not record the clinical outcomes were excluded. Other exclusion criteria were annotation of embryo grading or morphokinetical parameters from embryologists. Moreover, the observation that the following pairs of studies were part of the same project led to the decision to include only the data from the full-length manuscript. Tran’s and Ueno’s studies were part of the same project [21,26]. Thus, only Ueno’s study was included initially, as it presented the full dataset [26].

### 2.4. Data Extraction

Two authors performed the data extraction independently, based on the selection criteria, and a senior author reviewed the data analysis. Data on both the above-mentioned outcomes and information on the classification and the method employed for the development of the AI-model were extracted. Regarding Ueno’s study, since the authors do not provide their employed cut-off value or metrics other than the AUC, the excellent and good embryos as predicted by the model are herein considered as a positive prediction, whereas the fair- and poor-quality embryos are regarded as negative predictions [26].

### 2.5. Outcomes

The reported primary outcomes of interest were the rates of live birth and pregnancy loss rates. The rate of live birth was defined as the number of deliveries that resulted in at least one live birth and the pregnancy loss rate as the outcome of a clinical pregnancy that did not result in live birth. However, no study reported results for pregnancy loss rate outcome. The reported secondary outcomes were clinical pregnancy, clinical pregnancy with fetal heartbeat, and ploidy prediction. Clinical pregnancy was evaluated by ultrasonographic visualization of one or more gestational sacs or definitive clinical signs of pregnancy.

### 2.6. Bias Assessment

The risk of bias was assessed by two authors independently, employing the PROBLAST tool. Any disagreements were mediated by a senior author.

### 2.7. Deviations from Protocol

In the protocol, it was mentioned that the QADAS-2 tool would be employed for the risk of bias assessment. However, the PROBLAST tool was preferred as it is more suitable for studies presenting predictive models. The ploidy status prediction outcome was added.

#### Metrics and Measures

The metrics and measures employed in the present data synthesis are sensitivity, specificity, positive predictive value (PPV), negative predictive value (NPV), diagnostic odds ratio (DOR), Area Under the Curve (AUC) of the Summary of the Receiver Operating Characteristic (SROC), and the Observed:Expected (O:E) ratio. All the above mentioned metrics were calculated from the true-positive (TP), false-positive (FP), true-negative (TN), and false-negative (FN) predictions. Sensitivity is defined as the ability of an evaluation method to detect a true positive and is calculated as: TP/(TP + FN). Specificity is the ability of an evaluation method to detect a true negative and is calculated as TN/(TN + FP). PPV is defined as the probability that a positive prediction is indeed positive and is calculated as: TP/(TP + FP). NPV is defined as the probability that a negative prediction is indeed negative and is calculated as: TN/(TN + FN). The DOR of an evaluation method is the ratio of the odds of true positivity relative to the odds of false positivity. DOR is calculated as sensitivity × (1-sensitivity) × (1-specificity) × specificity. The SROC graph is conceptually very similar to the ROC. However, each data point comes from a different study, not a different threshold [27]. The AUC of the SROC was assessed similarly to the evaluation of the AUC of the ROC [28]. To further our analysis, the partial AUC (pAUC) was employed. The partial AUC considers only those regions of the ROC space where data have been observed [29]. The AUC as well as the pAUC may range from 0.5, corresponding to a random prediction, to 1, corresponding to an “always correct”/perfect discrimination prediction. Heterogeneity of the effect was evaluated through I^2^. Finally, the Observed:Expected (O:E) ratio was evaluated. The O:E ratio, when evaluating prediction models, is indicative of the models’ calibration [30,31]. The calibration of the model refers to its accuracy [32]. The positive outcomes are considered to be the “Observed”, in other words the numerator in the ratio, while the predicted positive outcomes serve as the “Expected”, i.e., the denominator. An O:E ratio lower than 1 presents an overestimation of the positive outcome, while an O:E ratio higher than 1 presents and underestimation of the positive outcome [31].

### 2.8. Statistical Analysis

Following the data extraction, a statistical analysis was performed. Sensitivity, specificity, positive and negative predictive value, diagnostic odds ratio, and Summary of the Receiver Operating Characteristic (SROC) of prediction of live-birth and clinical pregnancy were evaluated. Measure of effects entailed a univariate analysis with the employment of the random-effects model. Moreover, a bivariate analysis to plot the SROC curve was conducted. Since the heterogeneity between the studies was significantly high, the Bayesian approach was preferred, as frequentist estimation methods sometimes fail to produce reliable confidence intervals. Moreover, the inclusion of studies with different sample sizes, as well as the limited number of studies regarding the live-birth outcome could be better evaluated in a Bayesian estimation framework, which more naturally accounts for all parameter uncertainty in the derivation of credibility and probability intervals [30]. A subgroup analysis was performed according to the type of input, namely static images and time-lapse. The R programming language for statistical purposes was employed, and specifically the packages “metafor”, “mada”, and “metamisc”.

## 3. Results

The characteristics of each study included in the systematic review and data synthesis are presented in Table 1. As presented herein, 11 studies employed videos from time-lapse microscopy and/or morphokinetic data, 6 studies employed blastocyst static images, and 1 study evaluated both time-lapse microscopy and static blastocyst images. The majority of studies employed CNN for the model’s development. The model optimization differed significantly between the studies. While a number of studies opted for a k-times validation, others performed optimization by validating for k epochs. In k-times validation, the dataset is divided into k-stratified subsets, and in each validation process, one subset is employed as the validation (or test) dataset, whereas the other subsets are employed for training. When employing k epochs, the validation and the training subset remain constant, while the algorithm is optimized for k times, employing the specific subsets, representing the full dataset.

The majority of studies reported on a single outcome. However, a limited number of studies reported on two outcomes. More specifically the assessment of bias was performed according to the PROBLAST tool and is presented in Table 2.

All studies, with the exception of the study by Aparicio Ruiz and colleagues, were of retrospective nature. This is anticipated, as the studies reported on the development of novel prediction models. Kan-Tor et al.’s and Liao et al.’s studies reported on prediction of blastocyst formation, while the study by Liao and colleagues also reported on prediction of blastocyst quality. Due to the small number of studies reporting on other outcomes, a data synthesis could not be performed. AI seems to enable accurate prediction of blastocyst formation with an AUC between 0.75 and 0.83. The prediction of blastocyst quality similarly appears to be accurate since it employs AI prediction models with an AUC of 0.79. From the included studies, only the study by Ver Mileya and colleagues presented a performance comparison between AI and embryologists reporting on clinical outcomes. The study by Liao et al., further performed a performance comparison between AI and embryologists on blastocyst formation prediction. Both studies reported enhanced predictive capabilities when employing AI. When examining blastocyst formation prediction, a significant difference was presented in the predictive capability of AI versus the embryologists. However, in comparing clinical outcomes, it appears that AI has still some way to go prior to claiming to significantly surpass clinical embryologists’ predictive competence when it comes to primary outcome measures.

### 3.1. Prediction of Live-Birth

A total of four studies with five arms reported results regarding prediction of live birth. A total of 1981 embryos were transferred, resulting in 578 live births. There were 244 true positive predictions, 197 false positives, 1300 true negatives, and 334 false negatives. Sensitivity, specificity, PPV, NPV, and DOR, as well as the heterogeneity of each measure, are presented in Figure 2. When evaluating the SROC, the AUC was found to be 0.905, while the partial AUC (pAUC) was 0.755 (Figure 3). Employing the Bayesian approach, the total Observed:Expected ratio (O:E) was 1.12 (95%CI: 0.26–2.37; 95%PI: 0.02–6.54).

### 3.2. Sensitivity Analysis on Live Birth Prediction

When excluding studies that presented as conference abstracts, only, two out of the four identified studies were eligible for inclusion considering the live birth outcome. Sensitivity was 32.7% (95%CI: 10.2–67.5%), specificity 85.5% (95%CI: 57.7–96.2%), PPV 39.3% (95%CI: 26.1–54.3%), and NPV 81.1% (95%CI: 70.2–88.6%). The AUC was 0.665. When evaluating on bias assessment, only Sawada’s study was found to be eligible for inclusion, and thus data synthesis was not performed.

### 3.3. Secondary Outcome Measures

#### 3.3.1. Prediction of Pregnancy

A total of 10 studies with 10 arms reported results regarding prediction of clinical pregnancy. A total of 6794 embryos were transferred, resulting in 2765 clinical pregnancies. The true positive predictions were 2047, the false positives were 2108, the true negative were 2120, and the false negatives were 718. Sensitivity, specificity, PPV, NPV, and DOR, as well as the heterogeneity of each measure, are presented in Figure 4. When evaluating the SROC, the AUC was found to be 0.716, while the partial AUC (pAUC) was found to be 0.693 (Figure 5). According to the Bayesian approach, the total Observed:Expected ratio (O:E) was 0.92 (95%CI: 0.61–1.28; 95%PI: 0.13–2.43).

#### 3.3.2. Prediction of Clinical Pregnancy with Fetal Heart-Beat

A total of seven studies with seven arms reported results regarding the prediction of clinical pregnancy with fetal heartbeat. A total of 5828 embryos were transferred, resulting in 2255 clinical pregnancies with fetal heartbeat. There were 1708 true positive predictions, 2020 false positives, 1752 true negatives, and 547 false negatives. Sensitivity, specificity, PPV, NPV, and DOR, as well as the heterogeneity of each measure, are presented in Figure 6. When evaluating the SROC, the AUC was found to be 0.722, while the partial AUC (pAUC) was found to be 0.774 (Figure 7). According to the Bayesian approach, the total Observed:Expected ratio (O:E) was 0.77 (95%CI: 0.54–1.05; 95%PI: 0.21–1.62).

#### 3.3.3. Prediction of Ploidy Status

A total of four studies reported results regarding the prediction of embryo ploidy status. A total of 772 embryos were evaluated regarding their ploidy status. A total of 293 were assessed as euploid, and the remaining 479 were assessed as aneuploid. Sensitivity, specificity, PPV, NPV, and DOR, as well as the heterogeneity of each measure, are presented in Figure 8. When evaluating the SROC, the AUC was found to be 0.78, while the partial AUC (pAUC) was found to be 0.636 (Figure 9). According to the Bayesian approach, the total Observed:Expected ratio (O:E) was 2.05 (95%CI: 0.79–3.17; 95%PI: 0.02–6.47).

A summary of the results, including subgroup analysis, is presented in Table 3.

## 4. Discussion

The development of machine learning has led to the employment of AI in order to enhance clinical care [50]. Therefore, it is not surprising that several areas of medicine, including the field of reproductive medicine, have embraced the AI age [51,52,53]. AI is a noninvasive approach and has been applied in several fields of reproductive medicine, such as sperm morphology, automation of follicle count, automatic embryo cell stage prediction, embryo grading, and prediction of implantation potential, as well as development of improved stimulation protocols [15]. This innovative technology has been reported to contribute to improving embryo classification from subjective and morphology-based to automated and objective [54].

According to the results of our study, the employment of AI appears to hold significant promise for the future of IVF; however, data sourced herein present significantly high heterogeneity, compromising the level of certainty. The present prediction models should be subject to improvement prior to representing the optimal choice. The high heterogeneity observed in our results may be attributed to the different protocols employed for the development of their models. Regarding the prediction of live birth, it may be observed that the four prediction models present higher specificity than sensitivity, as well as a higher NPV compared to PPV. In interpreting the results on the NPV and the specificity metrics, it appears that the models are designed to accurately predict negative outcomes. This may be attributed to the fact that the live birth population represents less than 50% of the sample size, and a higher sample size for the negative outcome is thus provided. However, the confidence intervals are wide, and thus safe conclusions cannot yet be reached. The high diagnostic odds ratio means that a true-positive prediction is more probable than a false-positive, showcasing the accuracy of the prediction models. The prediction models seem to aptly estimate the prediction of live-birth, as the confidence interval of the O:E ratio includes the value of one. However, in analyzing the prediction intervals, a number of studies have either overestimated or underestimated the live-birth result. The AUC of the SROC at 0.905 is considered excellent; however, the pAUC of 0.755 is considered only of good prognostic value. The partial AUC has been proposed as an alternative measure to the full AUC. When using partial AUC, only the regions of the ROC that include observed data are considered, corresponding to the metrics of sensitivity or specificity [29]. The low partial AUC, in contrast to the full AUC, may be attributed to the fact that the study by Miyagi and colleagues presents a very low sensitivity and positive predictive value [35]. However, it should be mentioned that the study by Miyagi et al., includes the highest sample size, highlighting the statistical power that the study holds. This fact, along with the different designs of the prediction model by Miyagi et al., may be a reason behind the high heterogeneity observed [35].

Regarding the pregnancy rates, the predictive capabilities of the AI models seem to be less effective. In this case, sensitivity is higher than specificity, and NPV is higher than PPV. The fact that sensitivity and NPV are higher may indicate that the prediction models are designed to lower false-negative predictions. It seems that the majority of prediction models for clinical pregnancy have a different design than the models for live birth prediction, which seem to attempt to maximize true negative predictions. Similarly to prediction of live-birth, the high diagnostic odds ratio means that it is more probable that a true-positive prediction is more probable than a false-positive. The prediction models seem to correctly estimate the prediction of live birth, as the confidence interval of the O:E ratio includes the value of 1. However, as may be observed from the prediction intervals, a number of studies have either overestimated or underestimated the pregnancy result. The AUC and partial AUC are between 0.71 and 0.69, thus exhibiting a marginally good prediction capability. It should be noted that the results of the present study are more robust when evaluating prediction of pregnancy compared to evaluation of prediction of live birth. This may be due to the larger number of studies included.

The majority of studies have focused on predicting clinical pregnancy with fetal heartbeat. In this context, sensitivity is higher than specificity and NPV is higher than PPV. Similarly to prediction of live birth and prediction of pregnancy, the high diagnostic odds ratio means that a true-positive prediction is more probable than a false-positive. The prediction models seem to correctly estimate the prediction of live birth, as the confidence interval of the O:E ratio includes the value of one. However, as can be observed from the prediction intervals, a number of studies have overestimated the clinical pregnancy with the fetal heartbeat result. The AUC and partial AUC are between 0.72 and 0.77, thus exhibiting a good prediction capability. It should be noted that the results of the present study are more robust when evaluating prediction of clinical pregnancy with fetal heartbeat compared to evaluation of the other outcomes; this is due to the lower heterogeneity and narrower confidence intervals observed.

It should be mentioned that the prediction accuracy of the embryologist is regarded to be between 60–70% [22,39]. The AUC of the embryologists’ prediction is proposed to range between 0.63 and 0.70 [55,56]. Taking this into account, it may be claimed that the AI prediction models offer slightly improved prediction capabilities. However, it should be underlined that embryologists are more accustomed to providing factorial rather than dichotomous outcomes. Further, according to VerMileya and colleagues’ study, the majority of the embryologists’ ratings report on the grading classification grade of 3 in a five-grade scale. It appears that clinical embryology practitioners may avoid grading embryos as top- or poor-quality [22]. Thus, the true-positive and true-negative prediction values may be similar to the ones provided by AI when employing slightly different cut-off values. For AI-based prediction models to be relied on and employed in clinical practice, higher predictive capabilities are required.

Comparing time-lapse video with static images, safe conclusions cannot be drawn yet. The majority of studies employed time-lapse imaging. Especially regarding the prediction of live-birth and ploidy status, only a single study for each outcome employed static images. Thus, regarding these outcomes, safe conclusions regarding the optimal type of input may not yet be drawn. Evaluating prediction of clinical pregnancy, the two types of input do not seem to perform differently. As can be observed from Table 3, the only difference between the two types of outcomes on clinical pregnancy prediction is the slightly higher specificity when employing time-lapse. Regarding clinical pregnancy with fetal heartbeat, the two types of input seem to perform significantly differently. To elaborate, input of time-lapse seems to outperform static images regarding the metrics of sensitivity and PPV, whereas prediction models employing static images report enhanced NPV. However, this can be attributed to the different designs of the prediction models. According to literature, it remains unclear whether time-lapse imaging and the employment of morphokinetics can optimize embryo selection and thus enhance clinical outcomes [10]. Further studies are required in order to delineate the efficacy of time-lapse imaging both on embryo selection and as an input for AI-based prediction models.

In evaluating the prediction accuracy of the ploidy status, the predictive capabilities of AI models seem to be adequate. Specificity was higher than sensitivity, while NPV was higher than PPV. This may indicate that the prediction models are designed to primarily focus on lowering false aneuploid predictions rather than distinguishing the true euploid embryos. Employment of AI in models predicting euploidy status may present as a necessity in the not-so-distant future. A recent meta-analysis reported that PGT-A fails to improve live birth rates in the general population. On the other hand, it seems to be beneficial for specific patient groups, namely for women aged over 35 years old [57]. Therefore, it is important to employ AI-based prediction models that focus on indicating the embryos that may be at risk of aneuploidy, which should subsequently be subjected to biopsy. It may be of great clinical significance to develop a prediction model achieving a high PPV—even 100% if possible—in order to ascertain minimization of false euploid predictions. In this particular scenario, a high PPV is more significant than the other metrics, as the predicted euploid embryos would not be subjected to further unnecessary PGT-A evaluation, while the ones predicted to be aneuploid could be subjected to PGT-A for further analysis. The diagnostic odds ratio seems promising in this direction, as it suggests that a true-positive prediction is more probable than a false-positive. The prediction models seem to correctly estimate the prediction of euploidy, as the confidence interval of the O:E ratio includes the value of one. However, as it may be observed from the prediction intervals, a number of studies have either overestimated or underestimated the ploidy result. The AUC and the partial AUC are between 0.75 and 0.58, which highlights the fact that more studies should be performed prior to robustly concluding on the effectiveness of AI models on accurately predicting ploidy status.

When evaluating prediction models, several metrics may be employed. Undoubtedly, the AUC of the SROC and the O:E ratio are the most important ones. AUC in particular is unaffected by the heterogeneity observed in the metrics of sensitivity, specificity, PPV, NPV, and DOR. This is attributed to the fact that while the aforementioned metrics may be influenced by different cut-off values, AUC remains a constant. The AUC ranges from 0.5 to 1 and provides information regarding the predictive capabilities of a model, with 0.5 being completely random and 1 providing perfect prediction. When designing a model, developers may opt to enhance different metrics by altering the cut-off value. However, the AUC remains constant; thus, it may be considered as an objective metric. The O:E ratio provides information regarding the over- or under-estimation of the positive outcome. An O:E ratio lower than one presents an overestimation of the positive outcome, while an O:E ratio presents an underestimation of the positive outcome. When the confidence interval of the O:E ratio includes the value of one, the model correctly estimates the positive outcome. While these two metrics should be considered the most important, other metrics, such as sensitivity, specificity, PPV, and NPV provide useful information and should not be ignored.

The limited number of studies evaluating prediction of live-birth is the major limitation of the present study. Moreover, the different designs employed when developing the prediction models may be considered another limitation, leading to increased heterogeneity. The wide confidence interval in the O:E ratio regarding live-birth prediction may be considered a reason for caution when interpreting the results of the present study. Additionally, the threshold set regarding the study by Ueno et al., may have influenced the outcomes of sensitivity, specificity, PPV, and NPV. Furthermore, the inclusion of women regardless of their age may have influenced the results of the present data synthesis, as advanced maternal age (AMA) has been associated with lower clinical pregnancy and live birth rates. It may be possible that different designs for the prediction models may be required to analyze different populations. To elaborate on this, one could hypothesize that a model with a lower false negative rate is required for populations with a lower number of available embryos. A lower false negative rate will not lead to discarding embryos with a high implantation potential. On the other hand, a model with a lower false positive rate may be required for a “good-prognosis” population. A further reason for caution when interpreting results of the present study is the limited number of included studies that are characterized as having a low risk of bias. Due to this fact, a subset analysis including solely studies with an overall low risk of bias could not be performed. Moreover, when attempting subset analysis based on the items with bias, or excluding studies that presented at least two items with high risk of bias, the heterogeneity observed between the studies was higher. Thus, the authors opted to refrain from including the subset analysis in order to avoid introducing a lower level of certainty.

AI technology presents several limitations. In particular, neural networks suffer from overfitting during training. This can lead to neural networks providing inconsistent results. Furthermore, most neural networks are limited to imaging systems that were employed in the training process and are performed with reduced adaptive capacity to different imaging models [9]. Further research is required on the definition of the practical aspects when accommodating this approach. In the clinical setting, the majority of laboratories do not employ similar camera equipment as each other. This may result in reduced adaptability of the models in different clinical settings, which can cause significant problems for models employing static images. Regarding models that are based on time-lapse microscopy, it may be possible that different prediction models should be employed in different incubators.

On the other hand, AI employment may be viewed as challenging when considering ethics in the context of replacing human decision-making with machines. It has been noted that it may be difficult to explain to patients the results obtained by AI models in the medical context [58]. In the context of IVF, this may present with added challenges, especially taking into consideration the possibility of a futile IVF cycle. Furthermore, possible biases in datasets originating from specific population characteristics may negatively influence the predictive capabilities of the AI model [59]. In the future, AI may establish digital automation in the field of reproductive medicine, providing great advantages to infertile patients. In the meantime, we must acknowledge the current status, where AI still has some way to go to prove to be the gold standard approach, and while accepting its promising nature, at present, the embryologist still needs to opt for the safest and most effective practice. It may be beneficial to examine the scenario where the final decision of the embryologist relies upon the software for embryo classification [19].

In the IVF laboratory, it is the embryologist who decides on the tools to be employed. Does that dictate that the embryologist should have a thorough understanding of the characteristics of the model employed? Does delving into deep learning require deep understanding? When analyzing studies on models, the characteristics listed in Table 1 include outcome measure, sample size, type of input, neural network employed, and method for model optimization. It may be of value to acknowledge that although all embryologists may be largely familiar with outcome measure, sample size, and type of input, they may not be familiar with the type of neural network employed or the method for model optimization, which are technical aspects of the model development. Ideally, one should have a clear understanding of what each of these characteristics entail and how they differ in order to reach conclusions and make arguments about optimal practice. In future IVF laboratories, embryologists may be able to deeply comprehend these differences when researching the optimal AI model. Presently, it may be of value to bridge the gap between “deep knowledge” and “deep learning”.

The AI approach serves as a highly promising tool in the era of personalized medicine, a fact that is reflected in the increasing trend noted in funding AI research programs [52]. Further studies are required prior to total integration of AI models towards a universal clinical practice. The standardization of such processes requires broader and unbiased databases generated by the collaboration of several clinics in order to reduce data variability and avoid heterogeneous datasets [12,51,60]. Thus, the challenges associated with data confidentiality, along with the competition between clinics, should be effectively addressed to ascertain appropriate transparency insurance and intellectual property protection [60]. Moving towards this direction will enable sourcing the powerful data required to lead to robust results and solid conclusions. Fulfilling these requirements and producing concrete data may result in an extensive change in clinical practice. Could this mean that we are moving towards an implementation of a totally automated robotic IVF lab of the future? The intentions of the scientific community engaged in AI research and implementation are led by the shared goal of optimal practice. However, hitherto, AI models and time-lapse imaging may be regarded as an IVF add-on [61]. Further prospective studies and eventually randomized controlled trials (RCTs) and meta-analyses of RCTs should be conducted prior to including AI in clinical practice. It may be possible that a future IVF laboratory may include AI prediction models and omics technologies to be used for selecting the optimal embryo to transfer. In the meantime, the acquaintance of embryologists with novel computational approaches is essential in order to achieve optimal and cost-effective care for all patients and to prepare for future practice [19,51].

To conclude, this systematic review and data synthesis set out to answer the question: Can AI models provide robust predictions? The data sourced herein support the claim that AI can in fact provide accurate predictions. Taking this conclusion further and considering the “when and how” of implementation, the next step would be to provide an answer to the question “How do AI prediction models perform in comparison to morphology- and/or morphokinetics-based evaluation? Although AI performs with precision, these models have yet to prove their superiority compared to humans. Data from this systematic review indicate that although AI models appear marginally more effective, they still have some way before they can claim to significantly surpass clinical embryologists’ predictive competence. However, the lack of prospective studies and RCTs does not allow for a meta-analysis to describe this debate. As for where we go from here, it is a certainty that the standardization of procedures should be a prerequisite when developing a universally applicable AI-based prediction model. Including a higher number of centers and collaborations between different developers should enhance prediction capabilities, and this should be a prerequisite when moving towards AI becoming the gold standard.

## Figures and Tables

**Figure 1 biomedicines-10-00697-f001:**
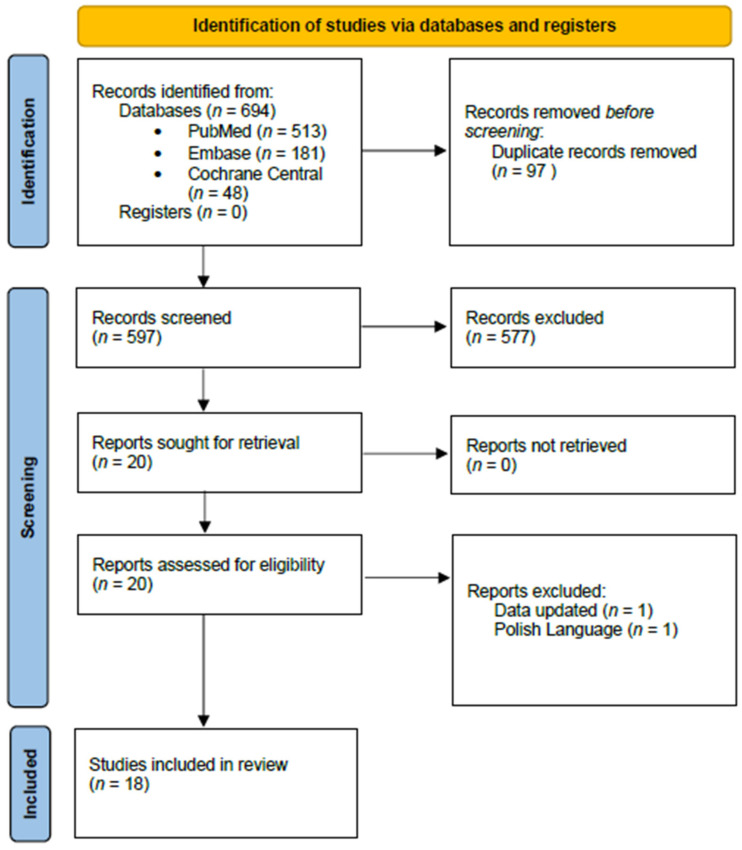
PRISMA flowchart.

**Figure 2 biomedicines-10-00697-f002:**
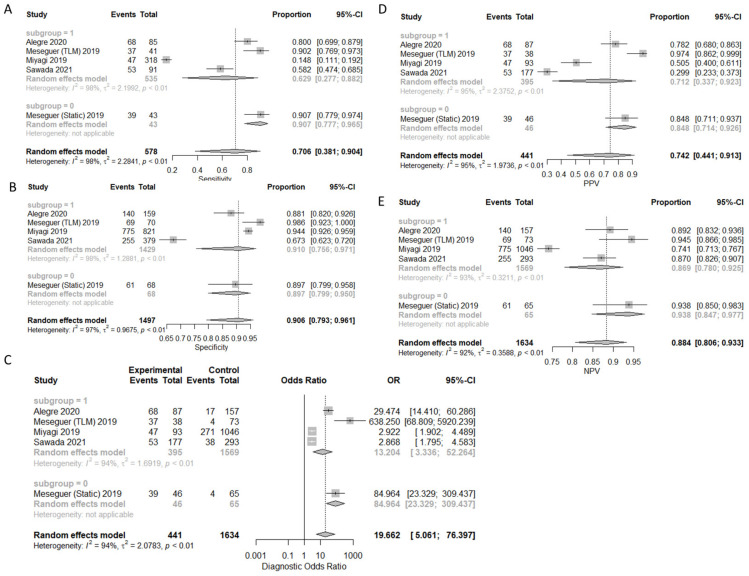
Forest plots representing: (**A**) sensitivity; (**B**). specificity; (**C**) DOR; (**D**) PPV; (**E**) NPV of the live birth prediction outcome. Subgroup “0” represents static images as the type of input, and subgroup “1” represents time-lapse.

**Figure 3 biomedicines-10-00697-f003:**
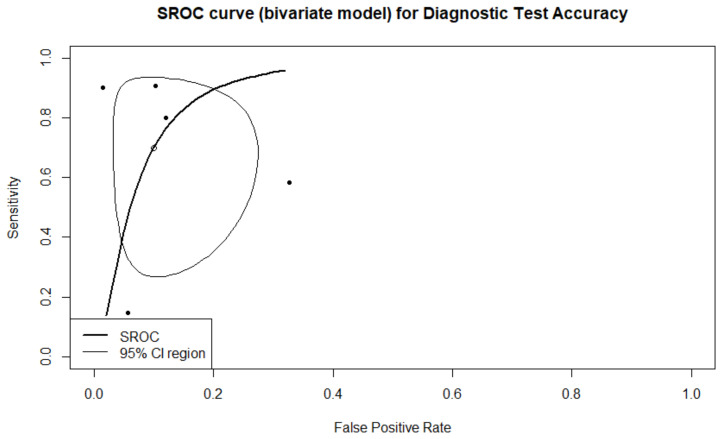
SROC of the live birth outcome.

**Figure 4 biomedicines-10-00697-f004:**
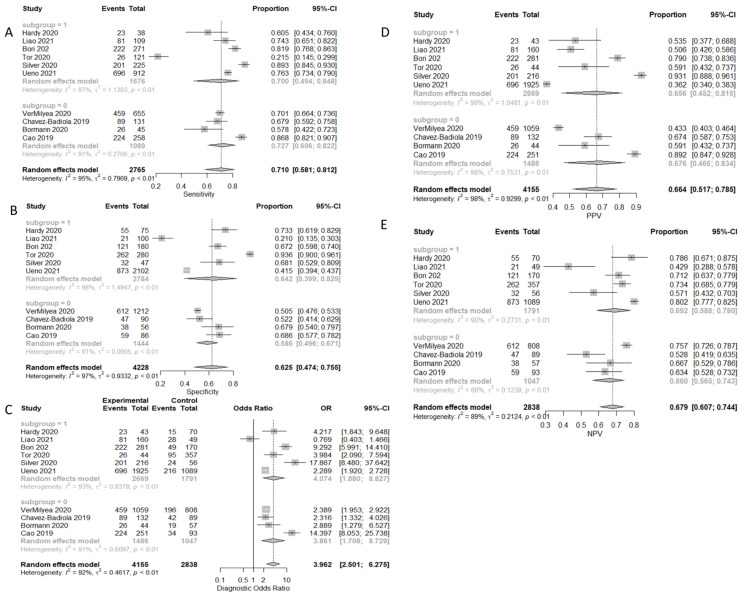
Forest plots representing: (**A**) sensitivity; (**B**) specificity; (**C**) DOR; (**D**) PPV; (**E**) NPV of the clinical pregnancy prediction outcome. Subgroup “0” represents static images as type of input, and subgroup “1” represents time-lapse.

**Figure 5 biomedicines-10-00697-f005:**
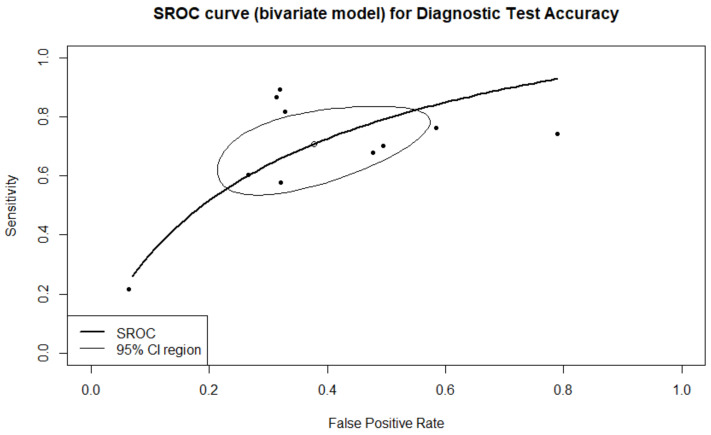
Prediction of pregnancy outcome.

**Figure 6 biomedicines-10-00697-f006:**
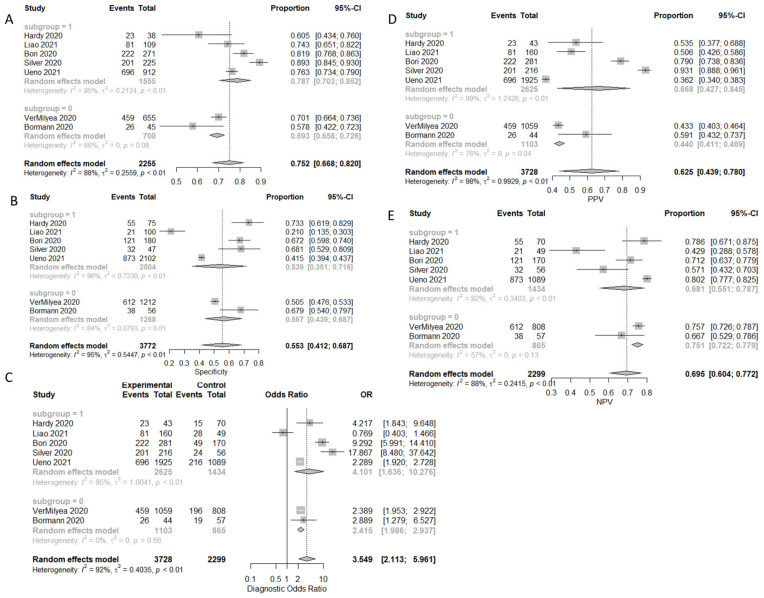
Forest plots representing: (**A**) sensitivity; (**B**) specificity; (**C**) DOR; (**D**) PPV; (**E**) NPV of the clinical pregnancy with fetal heart beat prediction outcome. Subgroup “0” represents static images as type of input, and subgroup “1” represents time-lapse.

**Figure 7 biomedicines-10-00697-f007:**
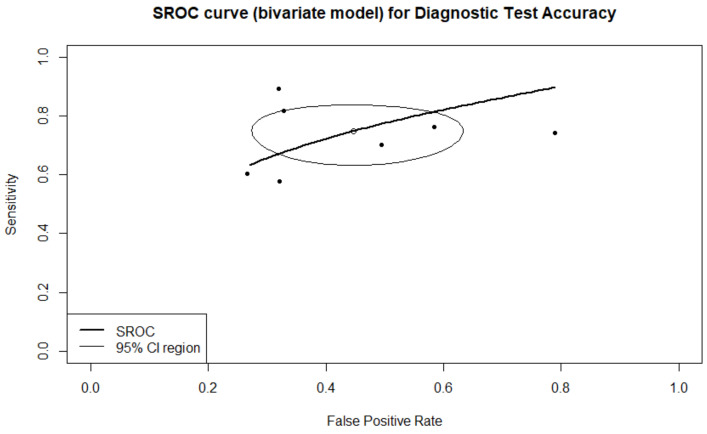
SROC of prediction of clinical pregnancy with fetal heartbeat.

**Figure 8 biomedicines-10-00697-f008:**
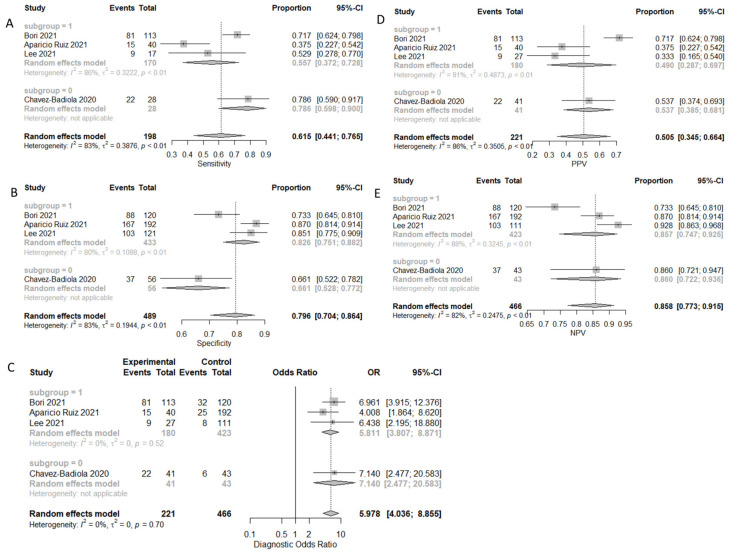
Forest plots representing: (**A**) sensitivity; (**B**) specificity; (**C**) DOR; (**D**) PPV; (**E**) NPV of the ploidy prediction outcome. Subgroup “0” represents static images as type of input, and subgroup “1” represents time-lapse.

**Figure 9 biomedicines-10-00697-f009:**
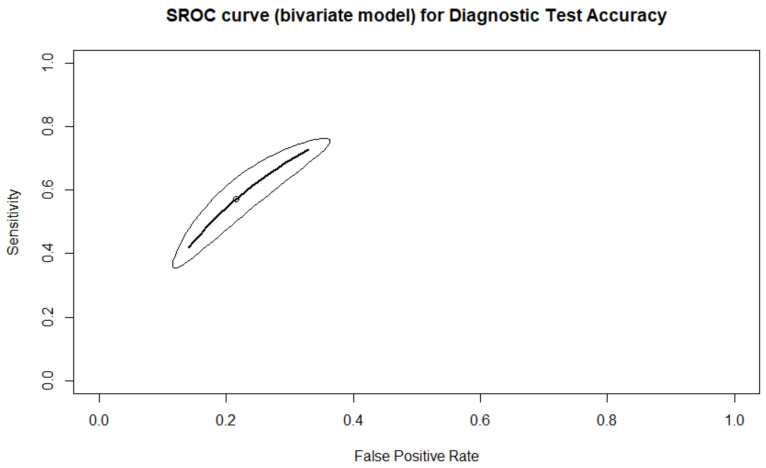
SROC of the prediction of ploidy status outcome.

**Table 1 biomedicines-10-00697-t001:** Study Characteristics.

Study	Outcome	Type of Input (TL/Static Images)	Sample Size	Type of AI Algorithm Employed	Model Optimization
Alegre 2020 [33]	Live-birth	TL	244	ANN	NP
Meseguer 2019 [34]	Live-birth	TL STATIC IMAGE	TL: 111 SI: 111	ANN	NP
Miyagi 2019 [35]	Live-birth	TL	1139	CNN	5-fold cross validation
Sawada 2021 [36]	Live-birth	TL	376	CNN with Attention Branch Network	Back-propagation for 5 epochs
Hardy 2020 [37]	Clinical Pregnancy with FHB	TL	113	CNN	NP
VerMilyea 2020 [22]	Clinical Pregnancy with FHB	STATIC IMAGES	1667	ResNet	Back-propagation and SGD for 5 epochs
Chavez-Badiola 2020 [38]	Clinical Pregnancy	STATIC IMAGES	221	SVM	10-fold cross validation
Liao 2021 [39]	Clinical Pregnancy with FHB	TL	209	DNN	NP
Bori 2020 [40]	Clinical Pregnancy with FHB	TL	451	ANN	5-fold cross validation
Kan-Tor 2020 [41]	Clinical Pregnancy	TL	401	DNN	20–60 epochs validation
Bormann 2020 [42]	Clinical Pregnancy with FHB	STATIC IMAGES	102	CNN	Genetic algorithm per 100 samples for a dataset of 3469 embryos
Silver 2020 [43]	Clinical Pregnancy with FHB	TL	272	CNN	NP
Cao 2018 [44]	Clinical Pregnancy	STATIC IMAGES	344	CNN	NP
Ueno 2021 [26]	Clinical Pregnancy with FHB	TL	3014	DNN	Back-propagation for 20 epochs and 5-fold cross validation
Bori 2021 [45]	Ploidy	TL	331	ANN	Back-propagation
Aparicio Ruiz 2021 [46]	Ploidy	TL	319	ANN	NP
Lee 2021 [47]	Ploidy	TL	138	CNN (3D ConvNets)	NP
Chavez-Badiola 2020 [48]	Ploidy	STATIC IMAGES	84	DNN	10-fold cross validation

TL: time-lapse microscopy; ANN: artificial neural network; CNN: convolutional network; DNN: deep neural network; ResNet: Residual Neural Networks; SGD: stochastic gradient resent with momentum; epoch: one complete pass through the entire dataset; NP: not provided.

**Table 2 biomedicines-10-00697-t002:** Assessment of Bias.

Study	Participants	Predictors	Outcomes	Analysis	Overall
Alegre 2020 [33]	-	+	+	+	-
Meseguer 2019 [34]	-	+	+	+	-
Miyagi 2019 [35]	+	+	+	-	-
Sawada 2021 [36]	+	+	+	+	+
Hardy 2020 [37]	-	+	+	+	-
VerMilyea 2020 [22]	+	-	+	+	-
Chavez-Badiola 2020 [38]	-	-	+	+	-
Liao 2021 [39]	-	+	+	+	-
Bori 2020 [40]	+	+	+	+	+
Kan-Tor 2020 [41]	+	+	+	+	+
Bormann 2020 [42]	-	-	+	+	-
Silver 2020 [43]	-	+	+	+	-
Cao 2018 [44]	+	-	+	+	-
Ueno 2021 [26]	+	+	+	-	-
Bori 2021 [49]	+	+	+	-	-
Aparicio Ruiz 2021 [46]	+	+	+	+	+
Lee 2021 [47]	-	+	+	+	-
Chavez-Badiola 2020 [48]	-	+	+	+	-

+: Low risk of bias; -: High risk of bias.

**Table 3 biomedicines-10-00697-t003:** Summary of the results.

Outcomes	Sensitivity	Specificity	PPV	NPV	DOR
Live-Birth	70.6% (38.1–90.4%)	90.6% (79.3–96.1%)	74.2% (44.1–91.3%)	88.4% (80.6–93.3%)	19.662 (5.061–76.397)
Live-Birth SI	90.7% (77.7–96.5%)	89.7% (79.9–95.0%)	84.8% (71.4–92.6%)	93.8% (84.7–97.7%)	84.964 (23.329–309.437)
Live-Birth TL	62.9% (27.7–88.2%)	91.0% (75.6–97.1%)	71.2% (33.7–92.3%)	86.9% (78.0–92.5%)	13.204 (3.336–52.264)
Clinical Pregnancy	71.0% (58.1–81.2%)	62.5% (47.4–75.5%)	66.4% (51.7–78.5%)	67.9% (60.7–74.4%)	3.962 (2.501–6.275)
Clinical Pregnancy SI	72.7% (60.6–82.2%)	58.6% (49.6–67.1%)	67.6% (46.6–83.4%)	66.0% (56.5–74.3%)	3.861 (1.708–8.729)
Clinical Pregnancy TL	70.0% (49.4–84.8%)	64.2% (39.9–82.9%)	65.6% (45.2–81.5%)	69.2% (58.8–78.0%)	4.074 (1.880–8.827)
Clinical Pregnancy with FHB	75.2% (66.8–82.0%)	55.3% (41.2–68.7%)	62.5% (43.9–78.0%)	69.5% (60.4–77.2%)	3.549 (2.113–5.961)
Clinical Pregnancy with FHB SI	69.3% (65.8–72.6%)	56.7% (43.9–68.7%)	44.0% (41.1–46.9%)	75.1% (72.2–77.9%)	2.415 (1.986–2.937)
Clinical Pregnancy with FHB TL	78.7% (70.3–85.2%)	53.9% (35.1–71.6%)	66.8% (42.7–84.5%)	68.1% (55.1–78.7%)	4.101 (1.636–10.276)
Ploidy	61.5% (44.1–76.5%)	79.6% (70.4–86.4%)	50.5% (34.5–68.1%)	85.8% (77.3–91.5%)	5.978 (4.036–8.855)
Ploidy TL	55.7% (37.2–72.8%)	82.6% (75.1–88.2%)	49.0% (28.7–69.7%)	85.7% (74.7–92.5%)	5.811 (3.807–8.871)
Ploidy SI	78.6% (59.0–90.0%)	66.1% (52.8–77.2%)	53.7% (38.5–68.1%)	86.0% (72.2–93.6%)	7.140 (2.477–20.583)

FHB: Fetal Heart Beat; SI: Static Image; TL: Time-Lapse.

## Data Availability

Not applicable.

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
