# Peer review of "Reporting on the Value of Artificial Intelligence in Predicting the Optimal Embryo for Transfer: A Systematic Review including Data Synthesis"

_biomedicines, 2022, doi:10.3390/biomedicines10030697_

Round 1

Reviewer 1 Report

The authors performed an extensive systematic review and meta-analysis dealing with very intriguing research question - value of AI for predicting the IVF outcome. Methodology used to conduct this systematic review seems ok and it raises interesting questions at the end of the discussion section. Although, I have to admit that I am not sure if the meta-analysis is a proper methodology for this study considering the high heterogeneity in study design of included studies, it might be better to make it as only data synthesis.

Few comments that authors should address:

Add rationale in the abstract on why it is relevant to perform such study.

Result section is very difficult to read. Please add forest plots in addition to SROC to help the reader understand the results better. Also, in my opinion it would be much simpler to reformat the results of four outcomes to a table - it would reduce the unnecessary repeated wording. For readers not acquainted with the technical terms it would be useful to describe what each of the measure means.

Page 8, sensitivity analysis: What does it mean "when evaluating only full-text articles" - this implies that the other 16 studies were not available in full text. Is that correct? If yes, please explain how did you perform all other analyses?

Have you tried to perform subset analyses based on the results of risk of bias assessment? In a way that you exclude the study that has high risk of bias (overall or per category).

Also, have you tried subset analysis with just specific AI algorithm or specific type of input (TL/static)?

Page 11, line 13: You state that the results of your study implies that AI has a significant improvement for IVF, based on what exactly do you conclude this from? Taking also into account that throughout the Results section you stated that heterogeneity between studies was extremely big.

It would be useful to provide more insights into the prediction models used in these 18 studies as well discuss in brief their study designs. Have you examined any additional parameters from each study from those that you report in Table 1? Did these studies comment or had analysed the performance of an embryologist?

In addition, some minor changes are needed:

  • missing space before/after brackets
  • check abbreviations whether they are mentioned in full text in first appearance
  • check for unnecessary spaces
  • page 5, lines 2-3, 8: references missing
  • check lines 46-48 if everything is correctly written, it feels like something is missing
  • page 12, lines 4, 7: references missing
  • please check other sites for possible missing reference

Author Response

The authors performed an extensive systematic review and meta-analysis dealing with very intriguing research question - value of AI for predicting the IVF outcome. Methodology used to conduct this systematic review seems ok and it raises interesting questions at the end of the discussion section. Although, I have to admit that I am not sure if the meta-analysis is a proper methodology for this study considering the high heterogeneity in study design of included studies, it might be better to make it as only data synthesis.

Dear Reviewer #1,

The authors are grateful for your time and effort in meticulously reviewing and improving the overall quality of the manuscript. Indeed, the heterogeneity observed is significantly high. Hence, following your aptly raised comment the authors have proceeded to employ the term data-synthesis throughout the manuscript, better describing the study. Indicatively, and as per your suggestion, please see the revised title, as well as throughout the text in page 3 lines 19, 43, page 6 line 42, page 19 line 53, page 21 line 21.

Few comments that authors should address:

Add rationale in the abstract on why it is relevant to perform such study.

Thank you for highlighting this omission on our end. The rationale of the study has now been added in the abstract. Please see revised abstract.

Result section is very difficult to read. Please add forest plots in addition to SROC to help the reader understand the results better. Also, in my opinion it would be much simpler to reformat the results of four outcomes to a table - it would reduce the unnecessary repeated wording. For readers not acquainted with the technical terms it would be useful to describe what each of the measure means.

Thank you for raising this. The forest plots for sensitivity, specificity, PPV, NPV and Diagnostic Odds Ratio for each of the four outcomes have now been added as per your suggestion. Please see figures 2,4,6,8. Further to this, a Table with the aforementioned outcomes is added to summarize our results and the unnecessary repeated wording has now been deleted respectively from all sections to provide a more reader-friendly format for the readership. Please refer to Table 3 as well as pages 9-17. Regarding the technical terms, following your comment, measures are now explained for the readership, in the measures subsection of  the Materials and Methods section. Please see pages 6-7 lines 32-10.

Page 8, sensitivity analysis: What does it mean "when evaluating only full-text articles" - this implies that the other 16 studies were not available in full text. Is that correct? If yes, please explain how did you perform all other analyses?

Thank you for highlighting the confusion that this description leads to. The sensitivity analysis included only the four studies reporting on the primary outcome, being live-birth prediction. Employing selection of “full-text articles”, the authors aimed to exclude manuscripts that presented in the form of conference abstracts. To be specific, two manuscripts were full-text published articles, and the other two were abstracts from the ASRM conference, published in the supplementary issue of the journal: “Fertility and Sterility”. As the EMBASE database was part of the searched databases, a number of abstracts, published in supplementary issues of scientific journals were included. In order to clarify this for the readership, this sentence has now been edited as follows: “When excluding studies that presented as conference abstracts, only two out of 4 identified studies were eligible for inclusion considering the live birth outcome. “ Please see page 10 lines 4-10

Have you tried to perform subset analyses based on the results of risk of bias assessment? In a way that you exclude the study that has high risk of bias (overall or per category).

Thank you for raising this. Due to the limited number of studies presenting with low overall risk of bias a subset analysis could not be performed. This is now mentioned in the limitations section of the manuscript. When attempting to perform the analysis based on single items or on studies that presented with at least two items with high risk of bias, the heterogeneity was even higher than when pooling all studies. Thus, the authors opted to refrain from including the analysis in the manuscript to avoid introducing a lower level of certainty. The fact that heterogeneity was higher when attempting subset analysis is mentioned in the limitations section of the study. Please refer to page 20 lines 6-13.

Also, have you tried subset analysis with just specific AI algorithm or specific type of input (TL/static)?

The authors appreciate your valuable comment. A subgroup-analysis on specific type of input has now been performed and respectively discussed in the discussion section. Please see the included forest plots, Table 3 and pages 18-19 lines 45-6.

Page 11, line 13: You state that the results of your study implies that AI has a significant improvement for IVF, based on what exactly do you conclude this from? Taking also into account that throughout the Results section you stated that heterogeneity between studies was extremely big.

Thank you for your comment. Indeed, the significantly high heterogeneity compromised the level of certainty. Thus, according to your suggestion, the statement has now been changed to: “According to the results of our study  employment of AI seems to hold significant promise for the future of IVF, however, data sourced herein presents with significantly high heterogeneity compromising the level of certainty” Please see revised version page 17 lines 13-16

It would be useful to provide more insights into the prediction models used in these 18 studies as well discuss in brief their study designs. Have you examined any additional parameters from each study from those that you report in Table 1? Did these studies comment or had analysed the performance of an embryologist?

Thank you for raising this. The design of the included studies has now been discussed in the study characteristics subsection of the manuscript. Moreover, a thorough discussion on additional parameters that were reported, but not included in the analysis has now been added. Two studies have reported on the performance of the embryologists, and this is now clearly stated in the manuscript. To analyze this further, all studies with the exception of the study by Aparicio Ruiz and colleagues were of retrospective nature. This is anticipated, as the studies reported on development of novel prediction models. Kan-Tor’s et al., and Liao’s et al.studies reported on prediction of blastocyst formation, while the study by Liao and colleagues reported also on prediction of blastocyst quality. Due to the small number of studies reporting on other outcomes, a data synthesis could not be performed. AI seems to enable accurate prediction of blastocyst formation with an AUC between 0.75 and 0.83. Prediction of blastocyst quality similarly appears to be accurate by employing AI prediction models with an AUC of 0.79.  From the included studies, only the study by Ver Mileya and colleagues presented a performance comparison between AI and embryologists, reporting on clinical outcomes. The study by Liao et al., further performed a performance comparison between AI and embryologists on blastocyst formation prediction. Both studies reported enhanced predictive capabilities when employing AI. When examining blastocyst formation prediction, a significant difference is presented on the predictive capability of AI versus the embryologists’. However, when comparing clinical outcomes, it appears that AI has still some way to go prior to claiming to significantly surpass the clinical embryologists ’predictive competence. Please see page 8 lines 9-25

In addition, some minor changes are needed:

  • missing space before/after brackets
  • check abbreviations whether they are mentioned in full text in first appearance
  • check for unnecessary spaces
  • page 5, lines 2-3, 8: references missing
  • check lines 46-48 if everything is correctly written, it feels like something is missing
  • page 12, lines 4, 7: references missing
  • please check other sites for possible missing reference

Thank you for your thorough and meticulous revision towards improving the overall performance of this manuscript. All minor changes that were suggested have now been addressed. Please see revised version page 1 line 38, page 5 lines 2-3,8, page 6 line 9, page 18 lines 1,4, page 19 line 53, page 21 line 14.

Reviewer 2 Report

The manuscript by Sfakianoudis et al, entitled “Reporting on the value of Artificial Intelligence in predicting the optimal embryo for transfer: A systematic review and meta-analysis", is an interesting systematic review investigating the predictive capabilities of Artificial Intelligence (AI) based prediction models regarding IVF outcome. The focus of this review is relevant to the field of reproduction, as the application of artificial intelligence in ART labs could be of great value.

The manuscript is well structured and the experimental design is appropriate.

The major limitation of this study is the limited number of studies included in the analyses, but the authors disclosed this limit in their discussion. The authors conclude that albeit AI performs with precision, nonetheless, these models have yet to prove their superiority when compared to the human element. Indeed, even if models appear marginally more effective, they still have some way before they can claim to significantly surpass the clinical embryologists ’predictive competence.

Author Response

The manuscript by Sfakianoudis et al, entitled “Reporting on the value of Artificial Intelligence in predicting the optimal embryo for transfer: A systematic review and meta-analysis", is an interesting systematic review investigating the predictive capabilities of Artificial Intelligence (AI) based prediction models regarding IVF outcome. The focus of this review is relevant to the field of reproduction, as the application of artificial intelligence in ART labs could be of great value.

The manuscript is well structured and the experimental design is appropriate.

The major limitation of this study is the limited number of studies included in the analyses, but the authors disclosed this limit in their discussion. The authors conclude that albeit AI performs with precision, nonetheless, these models have yet to prove their superiority when compared to the human element. Indeed, even if models appear marginally more effective, they still have some way before they can claim to significantly surpass the clinical embryologists ’predictive competence.

Dear Reviewer #2

Thank you for your aptly raised comment, indeed, the number of included studies is limited, and this is clearly stated in the limitation section of the study for the readership. The authors are grateful for your time and effort in reviewing this manuscript, and thank you for the kind and encouraging comments.

Round 2

Reviewer 1 Report

My suggestions have been followed by authors; this revised version of the manuscript is significantly improved and clearer than the initial version, and it could be accepted for publication.